# Ask To The Point: Open-Domain Entity-Centric Question Generation

**Yuxiang Liu    Jie Huang    Kevin Chen-Chuan Chang**
University of Illinois at Urbana-Champaign, USA
{yuxiang, jeffhj, kcchang}@illinois.edu

## Abstract

We introduce a new task called *entity-centric question generation* (ECQG), motivated by real-world applications such as topic-specific learning, assisted reading, and fact-checking. The task aims to generate questions from an entity perspective. To solve ECQG, we propose a coherent PLM-based framework GenCONE with two novel modules: content focusing and question verification. The content focusing module first identifies a focus as "what to ask" to form draft questions, and the question verification module refines the questions afterwards by verifying the answerability. We also construct a large-scale open-domain dataset from SQuAD to support this task. Our extensive experiments demonstrate that GenCONE significantly and consistently outperforms various baselines, and two modules are effective and complementary in generating high-quality questions.[1]

## 1 Introduction

Question generation (QG) aims to automatically generate questions from inputs such as raw texts (Du et al., 2017), knowledge bases (Bi et al., 2020), or images (Vedd et al., 2022). Particularly, text-based QG broadly benefits conversational chatbots to improve user interaction (Gao et al., 2019), educational materials to enhance reading comprehension (Wang et al., 2022), or QA dataset enrichment to boost QA development (Lyu et al., 2021). There are mainly two QG settings, answer-aware (Huang et al., 2021; Wu et al., 2022) and answer-agnostic (Back et al., 2021; Zhao et al., 2022), the difference between which is whether answers are known or not.

However, in many scenarios we care more about how to ask from an angle, i.e., from an *entity of interest* (EOI) perspective, rather than ask with an

answer or ask randomly, which we refer to as *entity-centric question generation* (ECQG). For example, in **topic-specific learning** (Liu et al., 2003), by generating questions focusing on a specified topic entity given a text, we can gain a better understanding of that subject. Second, in **assisted reading**, generating questions pertaining to a specific concept entity serves as reading anchors for efficient content digestion and information localization of desired knowledge (Yu et al., 2020). Third, in **fact checking**, generated questions targeting at different facts of EOI together with obtained answers can further form claims to be supported or refuted (Pan et al., 2021).

In this paper, we aim to solve ECQG, which is to generate an entity-centric question given a text and an EOI, emphasizing a particular aspect of the EOI. As answers are usually unknown, i.e., only the entity and its context are given in most scenarios, and unnecessary, i.e., entity alone suffices to locate answers, we define ECQG as answer-agnostic.

However, there are several challenges: (1) Lack of a dataset for ECQG. (2) Lack of centricity, as prior works treated input entities as answers (Sun et al., 2018; Fei et al., 2021; Wu et al., 2022) rather than as pivots to ask centered at them. (3) Lack of rationality, as existing answer-agnostic QG systems suffer from asking irrational, i.e., irrelevant or uninterpretable questions (Dugan et al., 2022). Summary-enhanced models (Zhou et al., 2021; Dugan et al., 2022) have been proposed to alleviate the issue, but they are domain-specific and only apply to detailed and in-depth input such as textbook or news articles, while for open-domain ECQG, where input texts vary in the level of detail, summaries do not always help. (4) Lack of answerability, as previous works tried to identify answer phrases to construct questions (Du and Cardie, 2017; Wang et al., 2019; Back et al., 2021), but such phrases are actually not treated as answers by the model, though it is a strong conditional restric-

---

[1] Code and dataset are publicly available at https://github.com/liuyuxiang512/ECQG.

tion for generation.

To address the lack of dataset, we construct a large-scale open-domain ECQG dataset from SQuAD (Rajpurkar et al., 2018). To further overcome the centricity, rationality, and answerability challenges, we design a novel **Gen**eration model with **C**ontent f**O**cusing and questio**N** v**E**rification (GenCONE), inspired by the human process of generating questions – humans tend to first identify a focus to form a draft question then verify the question afterward (Liu et al., 2020; Jhangiani et al., 2019).

Specifically, we propose *content focusing* (CF) and *question verification* (QV) modules, which are sequentially dependent with the main question generation (QG) module. Firstly, the upstream CF module identifies "what to ask", allowing the model to learn focus features as intermediate knowledge that bridges the entity and context, thereby improving question rationality. Secondly, the downstream QV module verifies questions through question answering, which imparts answerability-based knowledge into the model, thus improving question answerability. Thirdly, GenCONE jointly encodes entity and context and feeds them into CF, QG, and QV modules, which work together and enforce the model to learn entity-context relation to improve centricity.

Our main contributions are as follows: (1) We are the first to investigate *entity-centric question generation* (ECQG) problem. (2) We construct a large-scale open-domain dataset specific for ECQG and make it publicly available. (3) We propose a novel model called GenCONE, which is among the first works to build a coherent framework with both upstream and downstream sequentially dependent modules for answer-agnostic QG. (4) We conduct extensive experiments to demonstrate the superior performance of GenCONE and the effectiveness of its components.

## 2 Related Work

### 2.1 Question Generation

Question generation (QG) aims to automatically generate questions from raw texts (Du et al., 2017), knowledge bases (Bi et al., 2020), or images (Vedd et al., 2022). For text-based QG, there are mainly two settings, answer-aware (Huang et al., 2021; Wu et al., 2022) and answer-agnostic (Back et al., 2021; Zhao et al., 2022). The difference is whether answers are given or not. Previous works mostly assumed that answers exist and tried to capture answer-context relation with proximity-based (Sun et al., 2018), GNN-based (Fei et al., 2021), or structure-enhanced (Wu et al., 2022) models.

However, answers are not always known, and removing the constraints of answers increases the model's degrees of freedom, which is more beneficial for certain applications. Therefore, many researchers have been studying answer-agnostic QG since Du et al. (2017) first proposed it. Early works (Du et al., 2017; Scialom et al., 2019) targeted at totally uncontrolled QG, which introduces too much freedom and may generate irrelevant or uninterpretable questions. Some later works proposed to first identify question-worthy sentences (Du and Cardie, 2017) or phrases (Wang et al., 2019), and then generate questions conditioned on them; some other works (Wang et al., 2020; Back et al., 2021) proposed to incorporate answer span prediction or answer-containing sentence recovery to guide QG. A few recent works (Dugan et al., 2022; Zhao et al., 2022) also explored how human-written or machine-generated summaries help to improve the quality of generated questions. A recent work Reddy et al. (2022) focused on data augmentation for neural IR with QG conditioned on the sparsely attended words or phrases (entities) of the passage, where QG is application-specific, *i.e.*, QG for QA, and limited in entity types, *i.e.*, specified entity types are included. **To the best of our knowledge, there are no prior works studying open-domain ECQG**.

### 2.2 Entity-centric Text Generation

Existing entity-centric text generation works mainly focus on controllable summarization. Fan et al. (2018) is the first to bring forward controllable summarization considering control signals such as entities. They built it on a convolutional seq2seq model and used an anonymize-then-prepend method to enable entity-centric. He et al. (2020) later proposed keyword-controlled summarization based on pre-trained BART (Lewis et al., 2020a), and achieved entity-centric by treating entities as keywords. Liu and Chen (2021) proposed to control dialogue summarization flexibly with personal named entities to obtain personal-perspective dialogue summaries. These entity-centric summarization works usually prepended entities to text and applied seq2seq models without further investigation, assuming the seq2seq model itself can

learn the entity-context relation. Therefore, **how to fully investigate entity-context relation beyond vanilla seq2seq models has not been studied yet**.

## 2.3 Multi-Task Learning in Text Generation

Multi-task learning (MTL) is increasingly popular in text generation by training a model to perform multiple language tasks, where auxiliary tasks can be either sequentially dependent (Lewis et al., 2020b) or concurrent (Zhou et al., 2019a) with the main task, depending on whether input of a task is relying on output/hidden states of another task. Sequentially dependent auxiliary tasks are widely used in generation as either upstream (Lewis et al., 2020b) or downstream (Hosking and Riedel, 2019) tasks. In QG, Zhou et al. (2019b) introduced an upstream question type prediction task to generate more accurate interrogative words, while Zhang and Bansal (2019) used two downstream tasks, question paraphrasing and question answering, to address the "semantic drift" of questions. Particularly, for answer-agnostic QG, Wang et al. (2019) proposed an upstream question-worthy phrase extraction task to generate answerable questions, and Zhao et al. (2022) considered two upstream tasks, question type prediction and summarization, for event-centric educational QG. In this study, **we introduce both upstream and downstream modules, specifically investigating their integration and adaptation to a new task, with each module tailored to address specific challenges associated to ECQG.**

## 3 Method

We propose GenCONE, a PLM-based framework to handle the ECQG task with explicit and implicit guidance. In this section, we first give a formal definition of our problem and then dive into details of model design.

### 3.1 Problem Definition

The entity-centric question generation (ECQG) problem can be formulated as follows: given a text $T = \{t_1, t_2, \cdots, t_{|T|}\}$ and an entity of interest (EOI) $E = \{e_1, e_2, \cdots, e_{|E|}\}$, the objective is to generate a question $Q = \{q_1, q_2, \cdots, q_{|Q|}\}$ asking an aspect of entity $E$ from its context $T$. $t_i, e_j, q_k \in V$ are words in the context, entity, and question respectively; and $V$ is a vocabulary. The answer to the entity-centric question $Q$ is a text span that represents the specific aspect of EOI, such

as another entity related to it, excluding EOI itself. Our ECQG problem, by asking some aspect about an entity but the answer is not the entity itself, also differs from prior works that ask questions whose answer is the entity itself (Sun et al., 2018; Fei et al., 2021; Wu et al., 2022). We show an example of entity-centric question given context and entity in Table 1.

| Context: | **Beyonce** rose to fame in the late 1990s as lead singer of R&B girl-group Destiny's Child. |
|---|---|
| Entity: | Beyonce |
| Question: | When did Beyonce become popular? |

Table 1: An example. The central entity is **bold** in context and the underlined text span is the answer to the entity-centric question, which is an aspect of the entity and unknown.

### 3.2 GenCONE Model

The architecture of GenCONE is shown in Figure 1. Built on PLMs, the content focusing module (Section 3.2.1) first selects an entity-centric focus prior to generating questions, indicating "what to ask". Based on it, the question generation module (Section 3.2.2) learns a focus-aware context representation to generate question and its representation. Finally, the question verification module (Section 3.2.3) takes into account representations of both question and context to verify answers.

#### 3.2.1 Upstream: Content Focusing

Existing answer-agnostic QG systems suffer from asking irrational, i.e., irrelevant or uninterpretable questions (Dugan et al., 2022). To generate relevant and interpretable questions, we design an upstream *content focusing* (CF) module to plan for question generation by looking for "what to ask" related to EOI. By explicitly learning focus features, CF enables the model to "eavesdrop" (Zhang et al., 2022), i.e., obtaining these features through the learning of auxiliary task, and thus improve question rationality. Particularly, the focus features are exploited as intermediate knowledge bridging entity and context to interact with subsequent module.

**Encoder** GenCONE is built on a seq2seq backbone (Sutskever et al., 2014). We first use a pretrained Transformer encoder (Wolf et al., 2020) to jointly encode entity $E$ and text $T$. The input sequence is denoted as $C = \{x_1, x_2, \cdots, x_{|C|}\}$, where $C = E \langle sep \rangle T$ is a concatenation of entity

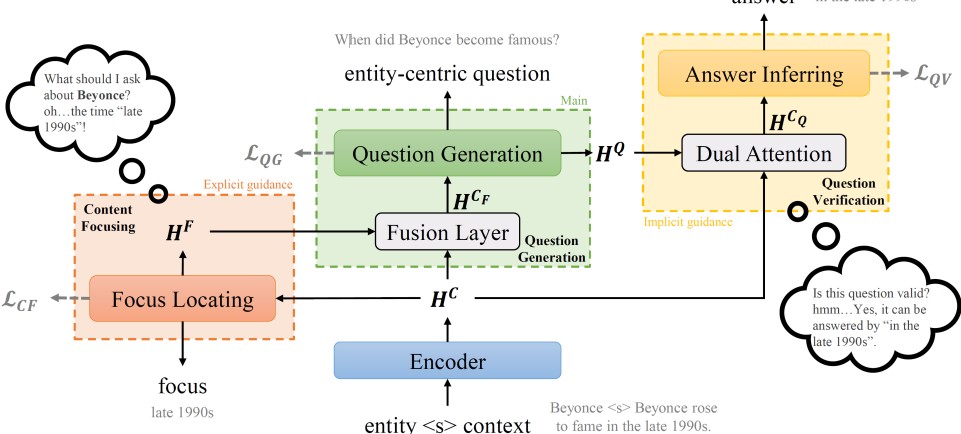

Figure 1: The overview of GenCONE architecture.

and text tokens separated with a special token, and $|C|$ is the length of input sequence. The obtained token-level input representation $\mathbf{H}^C$ is:

$$\mathbf{H}^C = \text{Encoder}(E\langle sep\rangle T) \in \mathbb{R}^{|C|\times d}, \quad (1)$$

where $d$ is the dimension for hidden representations and $\mathbf{H}_i^C$ is the $d$-dimensional representation for input token $x_i$. For simplicity, we set the hidden dimension of all modules the same as $d$.

**Focus Locating**   We consider content focus as a short text span. With token-level representation $\mathbf{H}^C$, we predict whether each token is a focus or not. Specifically, we use a pre-trained BERT (Devlin et al., 2019) to perform token classification:

$$\mathbf{H}^F = \text{BERT}(\mathbf{H}^C) \in \mathbb{R}^{|C|\times 2}. \quad (2)$$

The ground-truth focus $\mathbf{F} = [f_1 f_2 \cdots f_{|C|}]$ is a bit vector of the same length as input sequence, where each $f_i$ corresponds to an input token $x_i$, and $f_i = 1$ if $x_i$ belongs to the focus span. We treat the answer to the ground-truth question as content focus. The loss of CF is calculated as the cross-entropy loss between $\mathbf{H}^F$ and $\mathbf{F}$:

$$\mathcal{L}_{CF} = -\sum_{i=1}^{|C|} f_i \log(\mathbf{H}_i^F[0]). \quad (3)$$

### 3.2.2   Main: Question Generation

Question generation (QG) module is the main component to generate desired entity-centric questions, which is essentially a decoder, taking entity-centric context representation $\mathbf{H}^C \in \mathbb{R}^{|C|\times d}$ and focus features $\mathbf{H}^F \in \mathbb{R}^{|C|\times 2}$ as input.

**Fusion Layer**   We first fuse $\mathbf{H}^C$ and $\mathbf{H}^F$ to get a focus-aware context representation $\mathbf{H}^{C_F}$ as:

$$\mathbf{H}^{C_F} = [\mathbf{H}^C; \mathbf{H}^F]\mathbf{w}_{CF}, \quad (4)$$

where $[;]$ denotes concatenation along column axis and $\mathbf{w}_{CF} \in \mathbb{R}^{(d+2)\times d}$ is a linear transformation. Hence, we get the focus-aware context representation $\mathbf{H}^{C_F} \in \mathbb{R}^{|C|\times d}$ for subsequent decoding.

**Question Generation**   Taking $\mathbf{H}^{C_F}$ as input, we use a pre-trained Transformer decoder (Wolf et al., 2020) to generate a question, and we take the decoder's last hidden states $\mathbf{H}^Q = \text{Decoder}(\mathbf{H}^{C_F}) \in \mathbb{R}^{|Q|\times d}$ as question representation, where $|Q|$ is the length of the question sequence and $d$ is the dimension of hidden representations. Supposing $Q = \{q_1, q_2, \cdots q_m\}$ is the ground truth question, we calculate QG loss with teacher forcing as:

$$\mathbf{p}_j^Q, \mathbf{H}_j^Q = \text{Decoder}(\mathbf{H}^{C_F}, \mathbf{H}_{<j}^Q, q_{j-1}), \quad (5)$$

$$\mathcal{L}_{QG} = -\frac{1}{m}\sum_{j=1}^{m} \log \mathbf{p}_{j,q_j}^Q, \quad (6)$$

where $\mathbf{p}_j^Q$ is the probability distribution over decoding vocabulary at the $j$-th step, and $\mathbf{p}_{j,q_j}^Q$ is the probability of token $q_j$.

### 3.2.3   Downstream: Question Verification

To generate valid questions, previous answer-agnostic QG works (Du and Cardie, 2017; Wang et al., 2019; Back et al., 2021) proposed to identify answer phrases prior to generating questions. However, such extracted "answer" phrases are not treated as answers by their models, though it is

a strong conditional restriction for question generation. To ensure questions are answerable, it is infeasible to include an "answerability" feature when generating a question, as it will not be available as input at run time. Therefore, we design a downstream *question verification* (QV) module to examine answerability by inferring answers based on context and question. With such a verification step, QV is able to impart additional answerability-based knowledge into the model (Ruder, 2017), and thus improve question answerability.

**Dual Attention**   Taking $\mathbf{H}^C$ and $\mathbf{H}^Q$ as inputs, we first learn a question-aware context representation $\mathbf{H}^{CQ}$, which is inspired by Seo et al. (2016) to first fuse information bidirectionally, i.e., from $\mathbf{H}^C$ to $\mathbf{H}^Q$ and from $\mathbf{H}^Q$ to $\mathbf{H}^C$, and then unify both to get $\mathbf{H}^{CQ} \in \mathbb{R}^{|C| \times d}$.

Mathematically, we first calculate a similarity matrix $\mathbf{S} \in \mathbb{R}^{|C| \times |Q|}$, with each element $\mathbf{S}_{ij} = \alpha(\mathbf{H}_i^C, \mathbf{H}_j^Q)$, where $\mathbf{H}_i^C$ and $\mathbf{H}_j^Q$ are embeddings of the $i$-th context token and the $j$-th question token respectively. We use the same $\alpha(\mathbf{h}^c, \mathbf{h}^q) = \mathbf{w}_S^T[\mathbf{h}^c; \mathbf{h}^q; \mathbf{h}^c \circ \mathbf{h}^q]$ as in Seo et al. (2016), where $\mathbf{w}_S \in \mathbb{R}^{3d}$, $\circ$ is element-wise product, and $[;]$ is vector concatenation along column. We then derive attended embeddings as:

$$\mathbf{a}_i = \mathrm{softmax}(\mathbf{S}_{i,:}) \in \mathbb{R}^{|Q|},$$
$$\widetilde{\mathbf{H}}_i^Q = \sum_j \mathbf{a}_{ij} \mathbf{H}_j^Q \in \mathbb{R}^d,$$
$$\mathbf{b} = \mathrm{softmax}(\max_{row}(S)) \in \mathbb{R}^{|C|},$$
$$\widetilde{\mathbf{h}}^c = \sum_i \mathbf{b}_i \mathbf{H}_i^C \in \mathbb{R}^d,$$

where $\max_{row}$ is to perform the maximum function across row axis. Thus $\widetilde{\mathbf{H}}^Q \in \mathbb{R}^{|C| \times d}$ and we tile $\widetilde{\mathbf{h}}^c$ $|C|$ times to get matrix $\widetilde{\mathbf{H}}^C \in \mathbb{R}^{|C| \times d}$. We then obtain token representation $\mathbf{H}_i^{CQ} = \beta(\mathbf{H}_i^C, \widetilde{\mathbf{H}}_i^Q, \widetilde{\mathbf{H}}_i^C)$, where $\mathbf{H}_i^{CQ}$ is the $i$-th row vector corresponding to the $i$-th context token, $\beta$ is defined by $\beta(\mathbf{h}^c, \widetilde{\mathbf{h}}^q, \widetilde{\mathbf{h}}^c) = \mathbf{w}_{CQ}^T[\mathbf{h}^c; \widetilde{\mathbf{h}}^q; \mathbf{h}^c \circ \widetilde{\mathbf{h}}^q; \mathbf{h}^c \circ \widetilde{\mathbf{h}}^c]$, and $\mathbf{w}_{CQ} \in \mathbb{R}^{4d}$. Finally, we get the question-aware context representation $\mathbf{H}^{CQ} \in \mathbb{R}^{|C| \times d}$.

**Answer Inferring**   Answers are short text spans of input. After getting question-aware context representation $\mathbf{H}^{CQ}$, we use a pre-trained BERT (Devlin et al., 2019) to predict whether each token is answer or not, formally, $\mathbf{H}^A = \mathrm{BERT}(\mathbf{H}^{CQ}) \in \mathbb{R}^{|C| \times 2}$. The ground-truth answer is denoted as $\mathbf{A} = [a_1 a_2 \ldots a_{|C|}]$, which is a bit vector of the same length as input sequence, with $a_i = 1$ if corresponding input token $x_i$ is answer token and $a_i = 0$ if not. Similarly, the QV loss is the cross-entropy loss between $\mathbf{H}^A$ and $\mathbf{A}$:

$$\mathcal{L}_{QV} = -\sum_{i=1}^{|C|} a_i \log(\mathbf{H}_i^A[0]). \qquad (7)$$

### 3.2.4 Training Objective

We jointly train three modules end-to-end with a combined training objective as follows:

$$\mathcal{L} = \mathcal{L}_{QG} + \lambda_1 \mathcal{L}_{CF} + \lambda_2 \mathcal{L}_{QV}, \qquad (8)$$

where $0 < \lambda_1, \lambda_2 < 1$ control the relative importance of each associated loss. The CF loss enables the model to explicitly learn a content focus first and produce relevant and interpretable questions; while the QV loss allows answerability-based knowledge to be imparted into the model implicitly, and thus generating valid and answerable questions.

## 4  Experiment Setup

### 4.1  Dataset

We construct an ECQG dataset from SQuAD (Rajpurkar et al., 2018), an open-domain reading comprehension dataset originated from Wikipedia articles. Specifically, we use SQuAD v2.0[2], which has around 130k training samples and 12k testing samples. We first remove samples without answers so that all remaining questions are answerable. For samples where multiple answers exist for the same question, we vote for the answer to ensure answer quality, i.e., selecting the answer with the highest frequency, thus prioritizing the most commonly agreed-upon answer.

The key to construct an ECQG dataset is to obtain entity attribute of each sample. On the one hand, as Wikipedia titles are the core entities discussed in texts, we consider them as central entities if corresponding questions contain the title entities, assuming that they are the central entities of both contexts and questions. On the other hand, questions may not relate to title entities. In this case, we first use spaCy[3] to extract entities from contexts and questions respectively. If both context and question share and only share a common entity, this entity will be treated as the central entity. This

[2]https://rajpurkar.github.io/SQuAD-explorer/
[3]https://spacy.io/

| Split | Size | Entity Length: Mean(Min/Max) | Context Length: Mean(Min/Max) |
|---|---|---|---|
| Training | 42,128 | 1.74 (1/8) | 119.19 (20/653) |
| Validation | 3,364 | 1.88 (1/7) | 119.11 (20/445) |
| Testing | 2,338 | 1.94 (1/8) | 126.48 (25/540) |

Table 2: Statistics of our ECQG dataset.

| | good | bad |
|---|---|---|
| Training | 28 (93.33%) | 2 (6.67%) |
| Testing | 18 (90%) | 2 (10%) |

Table 3: Quality evaluation results of our dataset.

is to reduce the noise introduced by entity extraction, so that questions are tightly tied to a single key entity and more central to this entity. By filtering out samples that do not satisfy the above conditions and splitting the training set into training and validation sets, we finally get the dataset. The statistics of our ECQG dataset are in Table 2.

We conducted a manual analysis of the dataset we constructed. A sample is considered 'good' if the extracted entity is meaningful and the question is centered around this entity, such as when another semantically related entity serves as the answer. Conversely, a sample is labeled 'bad' if the question does not directly pertain to the extracted entity or if the extracted entity is non-specific, merely representing a general word or phrase. For this evaluation, we randomly selected 30 samples from the training set and 20 samples from the testing set. The results are presented in Table 3.

We further examined the erroneous samples identified in the dataset, with two representative examples shown in Table 4. A recurrent issue in these samples is the need to consider longer phrases encompassing the extracted "entity" as a whole. For instance, in Example 1, "the French House of Guise" should be treated as a single entity rather than just "Guise". Similarly, in Example 2, the appropriate entity is "school of Public Health" instead of merely "Public Health". This limitation stems from the constraints of the entity extraction tool we utilized. However, it is important to note that a significant portion of the dataset, exceeding 90%, remains accurate.

## 4.2 Implementation Details

We built GenCONE on PLMs T5 (Raffel et al., 2020) or BART (Lewis et al., 2020a). For each model, we experimented with its *base* and *large*

| |
|---|
| *Example 1:* |
| Question: "What name was given to the plot to usurp power from the French House of Guise?" |
| Entity: "Guise" |
| *Example 2:* |
| Question: "Where are the Harvard medical, Dental and school of Public Health located?" |
| Entity: "Public Health" |

Table 4: Examples of bad samples in ECQG dataset.

versions. CF and QV modules are based on pre-trained BERT (Devlin et al., 2019), with hidden dimensions the same as T5 or BART, i.e., if T5/BART is *base/large* version, BERT is *base/large* version as well. We implemented GenCONE in PyTorch 1.13, and experimented on NVIDIA A40 with 45G memory. We used the AdamW optimizer and set the weight decay = 0.01, maximum source length = 128, maximum target length = 32. For *base* versions, we set batch size = 64 and epoch = 15. We set early stop training if there were no better results for 3 epochs. For *large* versions, we set batch size = 32 and epoch = 10. We set $\gamma_1 + \gamma_2 = 0.3$. We tried learning rates in {1e-5, 2e-5, 5e-5, 1e-4} and selected the one with best validation results for different models. We ran models with different seeds and calculated the average metric scores.

## 4.3 Evaluation Metrics

We adopted three automatic metrics, BLEU, METEOR, and ROUGE (Lin and Och, 2004), which are widely used in previous QG works to evaluate the quality of machine-generated texts.

## 5 Experiment Results

### 5.1 Comparison with Existing Models

#### 5.1.1 Baselines

ECQG is introduced as a novel task within the answer-agnostic paradigm. For benchmarking ECQG and assessing our proposed method, we adapt several existing answer-agnostic QG models as baselines. We also benchmark ECQG with large language models (LLMs). For the adapted QG baselines, we prepend entities to contexts and apply fine-tuning; for LLMs, we utilize a few-shot prompting strategy.

**SummQG** (Dugan et al., 2022) is a QG model enhanced by text summarization. To adapt SummQG for ECQG, it is imperative to produce entity-centric summaries. Given the absence of definitive entity-centric summaries, we employed an entity-

centric summarizer, CTRLsum (He et al., 2020) – pre-trained on CNN/DailyMail– to generate these summaries. For the actual question generation process, we leveraged the QG model provided by Dugan et al. (2022). Due to the unavailability of entity-centric summaries for training, we kept the summarization component fixed, while evaluating both *pre-trained-only* and *fine-tuned* QG modules. Here, fine-tuning was achieved using the generated summaries paired with their corresponding ground truth questions.

**D-S-DRIL** (Zhou et al., 2021) is a BART-based model with an intermediate summarization step but sampling summaries and reconstructing questions exclusively based on the hidden states of the summary decoder. We used the model from Demszky et al. (2018) to convert QA pairs into declarative sentences. These were treated as entity-centric summaries and combined with ground truth questions for training. The summary generation and question generation processed were trained jointly, with $\lambda = 0.3$ as recommended by Zhou et al. (2021).

**TegTok** (Tan et al., 2022) is a knowledge-augmented encoder-decoder model. This model incorporates task-specific knowledge during encoding, and open-world knowledge during decoding. To ensure equitable comparison, we disregarded external knowledge, focusing solely on task-specific knowledge obtained from training data.

**GPT-4** (OpenAI, 2023), as a large language model, has showcased outstanding performance across a wide range of NLP tasks. It is notably adept in multimodal zero-shot, one-shot, and few-shot contexts. As a result, GPT-4 is also adopted as a benchmark to evaluate the ECQG dataset and to compare with other methods. We conducted in-context learning on GPT-4, using both 1-*shot* and 5-*shot* prompting techniques. Investigating in-context learning of GPT-4 across varied shots could offer more insights into the impact of demonstrations. Nevertheless, given that this is not the primary focus of our study and considering the cost of GPT-4 API, we limit our evaluation to two specific few-shot scenarios.

### 5.1.2 Results

The main results are presented in Table 5. For ECQG, GenCONE notably surpasses other answer-agnostic question generation models and LLMs in performance. In particular, it exceeds the performance of summarization-enhanced models like SummQG and D-S-DRIL, with an absolute gain

of 5% to 50%. For example, SummQG achieves a 31.27% $ROUGE_L$ score and D-S-DRIL records 43.28% $ROUGE_L$. In contrast, GenCONE attains a 46.12% $ROUGE_L$ score, marking a relative gain of 47.5% over SummQG and 6.6% over D-S-DRIL. This suggests that prior summarization-enhanced QG models may not be optimally suited for the ECQG task, aligning with our initial hypothesis. The knowledge-enhanced model, TegTok, posts a 42.39% $ROUGE_L$ score, which is a 11.12% improvement over the fine-tuned SummQG but still falls short of GenCONE by 3.73%. Furthermore, the automatic evaluation scores of most fine-tuned models surpass those of GPT-4. This is because fine-tuning allows these models to capture the inherent distribution of ECQG, hinting at significant potential to enhance GPT-4's domain adaptation to ECQG. Besides, despite a minimal performance discrepancy, GPT-4 with 5-shot prompting appears marginally less effective than its 1-shot counterpart, suggesting that increasing the shots from 1 to 5 may not enhance GPT-4's efficacy in ECQG. Overall, these findings validate that our proposed GenCONE is more adept at extracting knowledge from pre-trained models for ECQG than its contemporaries.

### 5.2 Comparison with Seq2Seq Models

#### 5.2.1 Baselines

To further evaluate GenCONE in terms of whether it better exploits the pre-trained Seq2Seq models, we experimented with different pre-trained Seq2Seq models, T5 (Raffel et al., 2020) and BART (Lewis et al., 2020a), and we tried both *base* and *large* versions. For all Seq2Seq models, we concatenate entity with context separated with a special token as input, and train using ground truth entity-centric questions.

#### 5.2.2 Results

The results in Table 6 show that GenCONE consistently performs better than vanilla Seq2Seq models. Across all settings, GenCONE scores better on all metrics compared with the corresponding vanilla Seq2Seq model. For example, based on $BART_{base}$, GenCONE improves Seq2Seq from 42.58% to 46.09%, with a relative gain of around 8.2%. These results further demonstrate that GenCONE can well exploit and improve significantly from pre-trained encoder-decoder models.

|            | BLEU-1 | BLEU-2 | BLEU-3 | BLEU-4 | METEOR | $\text{ROUGE}_L$ |
|------------|--------|--------|--------|--------|--------|--------|
| SummQG | 28.29 | 18.09 | 12.84 | 9.35 | 24.44 | 30.58 |
| $\text{SummQG}_{FT}$ | 29.67 | 18.54 | 11.95 | 11.75 | 25.03 | 31.27 |
| D-S-DRIL | 38.25 | 27.11 | 20.12 | 14.71 | 34.88 | 43.28 |
| TegTok | 37.45 | 24.41 | 17.39 | 12.48 | 32.95 | 42.39 |
| $\text{GPT-4}_{1\text{-}Shot}$ | 30.98 | 20.06 | 14.06 | 9.95 | 29.71 | 35.12 |
| $\text{GPT-4}_{5\text{-}Shot}$ | 30.49 | 19.59 | 13.70 | 9.74 | 29.22 | 34.50 |
| GenCONE | **40.21** | **29.45** | **22.40** | **16.98** | **37.74** | **46.12** |

Table 5: Comparison with QG models and LLMs. GenCONE here is built on $\text{T5}_{base}$. SummQG and $\text{SummQG}_{FT}$ denote *pre-trained-only* and *fine-tuned* models respectively. $\text{GPT-4}_{n\text{-}Shot}$ is GPT-4 with $n$-shot prompting.

|            | BLEU-1 | BLEU-2 | BLEU-3 | BLEU-4 | METEOR | $\text{ROUGE}_L$ |
|------------|--------|--------|--------|--------|--------|--------|
| $\text{T5}_{base}$ |  |  |  |  |  |  |
|     Seq2Seq | 38.60 | 27.31 | 20.15 | 14.76 | 35.08 | 43.27 |
|     GenCONE | **40.21** | **29.45** | **22.40** | **16.98** | **37.74** | **46.12** |
| $\text{T5}_{large}$ |  |  |  |  |  |  |
|     Seq2Seq | 37.66 | 26.82 | 19.92 | 14.70 | 34.69 | 43.74 |
|     GenCONE | **40.95** | **30.45** | **23.56** | **18.15** | **38.92** | **47.06** |
| $\text{BART}_{base}$ |  |  |  |  |  |  |
|     Seq2Seq | 36.83 | 27.07 | 20.45 | 15.43 | 35.70 | 42.58 |
|     GenCONE | **39.41** | **29.21** | **21.80** | **16.96** | **38.30** | **46.09** |
| $\text{BART}_{large}$ |  |  |  |  |  |  |
|     Seq2Seq | 36.52 | 26.82 | 20.29 | 15.35 | 35.19 | 43.62 |
|     GenCONE | **39.85** | **29.54** | **22.03** | **17.08** | **38.55** | **46.51** |

Table 6: Comparison with Seq2Seq models.

## 5.3 Ablation Study: Effect of CF/QV Modules

### 5.3.1 Baselines

To better understand the effectiveness of CF module and QV module, we conducted ablation study by removing either of them as model variants.

**GenCONE-CF** is a variant of GenCONE by removing QV module, which is only equipped with CF module. The loss is thus calculated by $\mathcal{L} = \mathcal{L}_{QG} + \lambda_1 \mathcal{L}_{CF}$.

**GenCONE-QV** is a variant of GenCONE by removing CF module, which is only equipped with QV module. Particularly, we set the focus-aware context representation the same as original context representation, i.e., $\mathbf{H}^{C_F} = \mathbf{H}^{C}$. The loss is calculated by $\mathcal{L} = \mathcal{L}_{QG} + \lambda_2 \mathcal{L}_{QV}$.

### 5.3.2 Results

The results are shown in Table 7. As we can see, either removing QV module (GenCONE-CF) or CF module (GenCONE-QV) results in a performance degradation, compared with the full model Gen-CONE, which shows that two modules are complementary to some degree. They can learn different knowledge and jointly contribute to improve the performance of GenCONE. When compared with Seq2Seq, either GenCONE-CF or GenCONE-QV consistently performs better, which also demon-strates that both content focusing loss and question verification loss helps to improve Seq2Seq significantly, and indicates the effectiveness of both CF and QV modules in GenCONE.

## 5.4 Human Evaluation

### 5.4.1 Evaluation Setup

In addition to machine evaluation, we also conducted human evaluation to evaluate the quality of generated questions. We focus on three aspects of question quality: entity centricity, rationality (relevance and interpretability), and answerability. We randomly selected 100 (entity, context, question) samples generated by the Seq2Seq model and GenCONE, as well as variants of GenCONE in Section 5.3, based on $\text{T5}_{large}$, and asked three students to evaluate four properties of generated questions. Students are required to answer: (1) **entity centricity**, whether the question is centered at the entity, i.e., asking an aspect related to entity from the context; (2) **relevance**, whether the question is semantically relevant to the context; (3) **interpretability**, whether the question makes sense in terms of context; (4) **answerability**, whether the question is answerable or not by the context. Each student is required to annotate agree(5), somewhat agree(4), neutral(3), somewhat disagree(2), or disagree(1). We then calculated average scores of

|              | BLEU-1 | BLEU-2 | BLEU-3 | BLEU-4 | METEOR | ROUGE$_L$ |
|--------------|--------|--------|--------|--------|--------|-----------|
| T5$_{base}$  |        |        |        |        |        |           |
| Seq2Seq      | 38.60  | 27.31  | 20.15  | 14.76  | 35.08  | 43.27     |
| GenCONE-CF   | 38.91  | 28.07  | 21.01  | 15.57  | 36.33  | 45.15     |
| GenCONE-QV   | 38.74  | 28.18  | 21.23  | 15.79  | 36.81  | 45.79     |
| GenCONE      | **40.21** | **29.45** | **22.40** | **16.98** | **37.74** | **46.12** |
| T5$_{large}$ |        |        |        |        |        |           |
| Seq2Seq      | 37.66  | 26.82  | 19.92  | 14.70  | 34.69  | 43.74     |
| GenCONE-CF   | 39.73  | 29.21  | 22.25  | 16.82  | 37.52  | 46.47     |
| GenCONE-QV   | 40.27  | 29.57  | 22.50  | 17.04  | 38.03  | 46.40     |
| GenCONE      | **40.95** | **30.45** | **23.56** | **18.15** | **38.92** | **47.06** |

Table 7: Ablation study: effect of CF/QV modules. Seq2Seq is the vanilla pre-trained encoder-decoder model T5, with *base* and *large* versions. GenCONE is our proposed full model with both CF and QV modules.

three students for all models.

### 5.4.2 Results

As shown in Table 8, our method surpasses the Seq2Seq model across all properties, indicating that GenCONE produces questions of superior centricity, rationality, and answerability. Notably, GenCONE significantly enhances question answerability. Both GenCONE variants display improvements over the Seq2Seq model: GenCONE-CF excels in rationality, while GenCONE-QV boosts answerability more effectively. Additionally, GenCONE and its variants augment entity centricity, highlighting the effectiveness of both modules in enhancing centricity. We hypothesize that the joint encoding of entity and context compels the model to discern the entity-context relationship, particularly through the integration of the main question generation module and two additional modules: content focusing and question verification. Human evaluations further underscore that our proposed GenCONE, equipped with content focusing and question verification modules, consistently crafts questions of a higher quality than those generated by Seq2Seq models.

|            | Cen. | Rel. | Int. | Ans. |
|------------|------|------|------|------|
| Seq2Seq    | 3.98 | 4.09 | 3.73 | 1.86 |
| GenCONE-CF | 4.07 | 4.20 | 3.81 | 2.32 |
| GenCONE-QV | 4.13 | 4.16 | 3.78 | 2.87 |
| GenCONE    | **4.21** | **4.24** | **3.86** | **3.05** |

Table 8: Human evaluation results of Seq2Seq and Gen-CONE. Cen., Rel., Int., and Ans. denote centricity, relevance, interpretability, and answerability respectively.

### 5.5 Case Study

To gain an insight of how content focusing and/or question verification perform for ECQG, we show three examples in Appendix A. In the first example, the question generated by Seq2Seq is general and irrelevant, and questions generated by Gen-CONE as well as its variants are more relevant in terms of context, which are asking more concretely. In the second example, questions generated by all models are relevant. However, the question generated by Seq2Seq is unanswerable from context, i.e., context is not sufficient to ensure it is answerable. In the third example, all models perform badly. Seq2Seq generates irrelevant questions while Gen-CONE generates unanswerable questions considering context. However, the question generated by GenCONE is more interpretable and makes sense. Therefore, compared with Seq2Seq, GenCONE can generate more relevant, interpretable, and answerable questions given context and entity. In addition, we further evaluated the results of GPT-3.5 on the ECQG dataset with zero-shot prompting. More details are explained in Appendix B.

## 6 Conclusion

We introduce a new task, entity-centric question generation (ECQG), motivated by realistic applications such as topic-specific learning, assisted reading, and fact checking. We also construct a large-scale open-domain ECQG dataset from SQuAD. To address rationality, answerability, and centricity issues of generated questions, we propose a coherent PLM-based framework called GenCONE and design two novel modules, content focusing and question verification. Experiment results, including both automatic and human evaluations, show that GenCONE significantly and consistently outperforms baselines in terms of automatic metrics and question quality including entity centricity, rationality, and answerability.

# Limitations

As we construct ECQG dataset from SQuAD, which contains factoid questions and answers are short text spans, our ECQG dataset inherits these characteristics. Therefore, we focus on factoid ECQG in this paper. Future works may investigate different types of questions, e.g., highly abstractive entity-centric questions. Besides, as answers are short text spans, we use token classification for extractive QA to infer answers. Otherwise, we will need to use abstractive QA modules instead, though the idea in this paper still applies. Lastly, our model introduces many parameters and requires sufficient GPU resources to train.

# Acknowledgements

This material is based upon work supported by the National Science Foundation IIS 16-19302 and IIS 16-33755, Zhejiang University ZJU Research 083650, IBM-Illinois Center for Cognitive Computing Systems Research (C3SR) and IBM-Illinois Discovery Accelerator Institute (IIDAI), grants from eBay and Microsoft Azure, UIUC OVCR CCIL Planning Grant 434S34, UIUC CSBS Small Grant 434C8U, and UIUC New Frontiers Initiative. Any opinions, findings, conclusions, or recommendations expressed in this publication are those of the author(s) and do not necessarily reflect the views of the funding agencies.

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

## A  Case Study of GenCONE

The examples are shown in Table 9.

## B  Case Study of LLMs

We further evaluated the performance of GPT-3.5 on the ECQG dataset with zero-shot prompting. By providing prompt "Please ask an entity-centric question for entity <entity tokens> from the passage and give the corresponding answer: <passage tokens>", we manually evaluated the outputs and compared them with ours and ground truth. We present examples below in Table 10.

In our analysis, we observed that GPT-3.5 tends to generate more intricate and open-ended questions compared with those generated by GenCONE or found in ground truth. Unlike the factoid questions predominant in our dataset, the questions generated by GPT-3.5 often require a more nuanced understanding of the context and typically cannot be answered by a text span of the input passage. Moreover, while some questions generated by GPT-3.5 can be answered within the given context, others extend beyond it, requiring additional information. Therefore, although GPT-3.5 can still generate questions that focus on specific entities and appear rational, GenCONE demonstrates superior performance in terms of answerability.

| | |
|---|---|
| *Example 1:* | |
| Context: | · · · The antigens expressed by tumors have several sources; some are derived from oncogenic viruses like human papillomavirus, which causes **cervical cancer**, while others are the organism's own proteins that occur at low levels in normal cells but reach high levels in tumor cells. · · · |
| Ground-truth: | What is the virus in humans that causes cervical cancer? |
| Seq2Seq: | What is cervical cancer? |
| GenCONE-CF: | What virus causes cervical cancer? |
| GenCONE-QV: | What virus causes cervical cancer? |
| GenCONE: | What virus causes cervical cancer? |
| *Example 2:* | |
| Context: | · · · In **southern California** there are also twelve cities with more than 200,000 residents and 34 cities over 100,000 in population. · · · |
| Ground-truth: | There are 34 cities in southern California that have a population exceeding what number? |
| Seq2Seq: | How many cities are in Southern California? |
| GenCONE-CF: | How many cities in southern California have more than 100,000 residents? |
| GenCONE-QV: | How many cities in southern California have more than 100,000 people? |
| GenCONE: | How many cities in Southern California have more than 200,000 residents? |
| *Example 3:* | |
| Context: | · · · Due to its energy content, O2 is used by complex forms of **life**, such as animals, in cellular respiration (see Biological role). Other aspects of O2 are covered in the remainder of this article. |
| Ground-truth: | What characteristic of oxygen makes it necessary to life? |
| Seq2Seq: | What is the most common form of life? |
| GenCONE-CF: | What is the energy content of the element O2? |
| GenCONE-QV: | What is the name of the complex form of life? |
| GenCONE: | What is the energy content of oxygen that is used by life? |

Table 9: Case study of GenCONE and its variants. GenCONE-CF and GenCONE-QV are model variants with only CF and QV modules respectively. All models are based on $T5_{large}$. Central entities are marked **bold**.

| | |
|---|---|
| *Example 1:* | |
| Entity & Context: | Lindzen's criticisms & IPCC author Richard Lindzen has made a number of criticisms of the TAR. Among his criticisms, Lindzen has stated that the WGI Summary for Policymakers (SPM) does not faithfully summarize the full WGI report . . . John Houghton, who was a co-chair of TAR WGI, has responded to **Lindzen's criticisms** of the SPM ... |
| GPT-3.5: | How did IPCC author Richard Lindzen's criticisms relate to the Third Assessment Report (TAR) and specifically to the WGI Summary for Policymakers (SPM)? |
| GenCONE: | Who has responded to Lindzen's criticisms of the WGI? |
| Ground-truth: | Who responded to Lindzen's criticisms? |
| *Example 2:* | |
| Entity & Context: | cervical cancer & Another important role of the immune system is to identify and eliminate tumors. This is called immune surveillance. The transformed cells of tumors express antigens that are not found on normal cells. To the immune system, these antigens appear foreign, and their presence causes immune cells to attack the transformed tumor cells. The antigens expressed by tumors have several sources; some are derived from oncogenic viruses like human papillomavirus, which causes **cervical cancer** ... |
| GPT-3.5: | How does the immune system recognize and respond to cervical cancer, and what is one of the sources of antigens associated with this type of cancer? |
| GenCONE: | What virus causes cervical cancer? |
| Ground-truth: | What is the virus in humans that causes cervical cancer? |
| *Example 3:* | |
| Entity & Context: | X.25 networks & There were two kinds of **X.25 networks**. Some such as DATAPAC and TRANSPAC were initially implemented with an X.25 external interface. Some older networks such as TELENET and TYMNET were modified to provide a X.25 host interface in addition to older host connection schemes . . . X.75 and X.121 allowed the interconnection of national **X.25 networks** ... |
| GPT-3.5: | How did the development of X.25 networks differ between various implementations, and how did X.75 and X.121 contribute to the interconnection of these networks? |
| GenCONE: | What were the two types of X.25 networks? |
| Ground-truth: | How many types of X.25 networks were there originally? |

Table 10: Case study of zero-shot GPT-3.5 and GenCONE. Central entities are also marked **bold** in its context.