# OpenReview forum: "Ask To The Point: Open-Domain Entity-Centric Question Generation"
_EMNLP/2023/Conference — EMNLP 2023 Findings_

### Official Review · Reviewer_GFDG · 2023-08-02

**Soundness:** 2

**Excitement:**

2: Mediocre: This paper makes marginal contributions (vs non-contemporaneous work), so I would rather not see it in the conference.

**Missing References:**

Some missing references, as the topic of generating multiple QA pairs from a single passage has been discussed in previous studies, such as:
- End-to-End Synthetic Data Generation for Domain Adaptation of Question Answering Systems: https://aclanthology.org/2020.emnlp-main.439/
- Unsupervised Question Answering via Answer Diversifying https://aclanthology.org/2022.coling-1.149/

**Paper Topic And Main Contributions:**

This paper focuses on the task named Entity-centric Question Generation (ECQG), an approach with two specifically designated modules is proposed to addredd this task. The proposed content focusing and question verification modules are employed to enhance the performance of QG systems for generating entity-centric questions. Experiments carried out on the ECQG dataset demonstrate the strong performance of the proposed approach.

Overall, while this research paper presents an intriguing task, the motivation, proposed approach, and experimental results are not entirely convincing. Firstly, the authors claim that ECQG is a new task, however, this argument is not entirely accurate. For example, the paper https://dl.acm.org/doi/10.1145/3477495.3531878 published in 2022 has already studied entity-centric question generation. Moreover, using NER tools to extract answer spans (including entities) for question generation has been investigated in previous studies such as Lewis et al., 2019, Lyu et al., 2021, etc. Therefore, this motivation may not be well-founded. Secondly, the proposed approach is not compelling, as ideas similar to the use of content focusing and question verification are widely used method in the QG field such as https://aclanthology.org/D17-1090.pdf and https://aclanthology.org/D18-1427/. More importantly, the experimental results are insufficient to demonstrate the effectiveness of the proposed approach. The authors only conducted intrinsic evaluation on one dataset, but coherence evaluation of QA pairs and human evaluation are necessary to examine the quality of the generated data. Additionally, extrinsic evaluation, such as using the synthetic data for QA, would provide additional insights into the practical utility of the proposed approach.









**Reasons To Accept:**

Strengths:
- This paper focuses on an interesting problem.
- The structure of this paper is clear and easy to follow.

**Reasons To Reject:**

- Firstly, the authors claim that ECQG is a new task, however, this argument is not entirely accurate. For example, the paper https://dl.acm.org/doi/10.1145/3477495.3531878 published in 2022 has already studied entity-centric question generation. Moreover, using NER tools to extract answer spans (including entities) for question generation has been investigated in previous studies such as Lewis et al., 2019, Lyu et al., 2021, etc. Therefore, this motivation may not be well-founded.

- Secondly, the proposed approach is not compelling, as ideas similar to the use of content focusing and question verification are widely used method in the QG field such as https://aclanthology.org/D17-1090.pdf and https://aclanthology.org/D18-1427/.

- More importantly, the experimental results are insufficient to demonstrate the effectiveness of the proposed approach. The authors only conducted intrinsic evaluation on one dataset, but coherence evaluation of QA pairs and human evaluation are necessary to examine the quality of the generated data. Additionally, extrinsic evaluation, such as using the synthetic data for QA, would provide additional insights into the practical utility of the proposed approach.

**Reproducibility:**

3: Could reproduce the results with some difficulty. The settings of parameters are underspecified or subjectively determined; the training/evaluation data are not widely available.

**Reviewer Confidence:**

3: Pretty sure, but there's a chance I missed something. Although I have a good feel for this area in general, I did not carefully check the paper's details, e.g., the math, experimental design, or novelty.

---

> ### Author Rebuttal · Authors · 2023-08-29
>
> ## Response to Reviewer GFDG
>
> We appreciate the reviewer's time and effort in providing feedback on our paper. We value the comments and concerns and would like to address them as follows.
>
> > Firstly, the authors claim that ECQG is a new task, however, this argument is not entirely accurate. For example, the paper https://dl.acm.org/doi/10.1145/3477495.3531878 published in 2022 has already studied entity-centric question generation.
>
> The given paper https://dl.acm.org/doi/10.1145/3477495.3531878 focuses on data augmentation for neural IR with question generation (QG) conditioned on the sparsely-attended words or phrases (entities) of the passage, where QG is application-specific (QG for QA) and limited in entity types (specified entity types are included).
>
> However, in our paper, motivated from realistic scenarios such as topic-specific learning, assisted reading, and fact checking, we propose ECQG as an independent new task, which is open-domain and generally applicable. We also emphasize entity-centric rather than entity-conditioned in our task, i.e, we ask from the entity-perspective and the answer to the entity-centric question Q is a text span that represents the specific aspect of the entity of interest (EOI), such as another entity related to it, excluding EOI itself. Thanks for providing the paper and we will involve this paper in our revised version and clarify the differences.
>
> > Moreover, using NER tools to extract answer spans (including entities) for question generation has been investigated in previous studies such as Lewis et al., 2019, Lyu et al., 2021, etc. Therefore, this motivation may not be well-founded.
>
> The paper by Lewis et al., 2019 is not cited in our references; and Lyu et al., 2021 employed NER to first tag entities in the summary sentence to facilitate the generation of appropriate question words in their summarization-informed question generation.
>
> In contrast, our approach does not utilize NER tools to extract answer spans. Instead, we introduce a coherent end-to-end model featuring two specialized modules: one for locating the focus and another for verifying answerability.
>
> > Secondly, the proposed approach is not compelling, as ideas similar to the use of content focusing and question verification are widely used method in the QG field such as https://aclanthology.org/D17-1090.pdf and https://aclanthology.org/D18-1427/.
>
> The first paper (https://aclanthology.org/D17-1090.pdf) introduced the concept of question pattern learning (e.g., "who founded # ?") in the field of QG for QA, and the second paper (https://aclanthology.org/D18-1427/) proposed to explicitly model interrogative words (question words) generation for answer-aware QG. None of them offer end-to-end models and there are no similar ideas with ours.
>
> Besides, in the QG field, though many auxiliary tasks have been proposed to enhance QG, especially answer-agnostic QG (Section 2.1 and 2.3), we are the first to propose the concepts of content focusing and question verification, and integrate both upstream and downstream auxiliary tasks for QG. Specifically, our implementation employs specialized methods within these modules optimized for ECQG.
>
> > More importantly, the experimental results are insufficient to demonstrate the effectiveness of the proposed approach. The authors only conducted intrinsic evaluation on one dataset, but coherence evaluation of QA pairs and human evaluation are necessary to examine the quality of the generated data. Additionally, extrinsic evaluation, such as using the synthetic data for QA, would provide additional insights into the practical utility of the proposed approach.
>
> We prioritize a generalized QG rather than focusing on specific use-cases like QA data enrichment. Thus we evaluated the quality of generated questions solely on our constructed dataset, aligning with most prior studies of QG (Section 2.1 and 2.3). We have conducted extensive experiments to show the outperformance of our model, including comparisons with adapted existing methods (Table 3), tests using different PLM backbones (Table 4), ablation studies to investigate the impact of model components (Table 5), and qualitative assessments through human evaluation (Table 6). While we appreciate the reviewer's suggestion to extend our experiments to include additional evaluations, such as using synthetic data for QA, we consider that to be beyond the scope of the current paper. We will explore this in our future research.
>
> > **Missing References**
>
> We thank the reviewer for providing additional references related to QG for QA pairs generation. We have included some relevant papers and will further incorporate these references into our revised version to provide a more comprehensive overview of related work.

---

### Official Review · Reviewer_G3ne · 2023-08-04

**Soundness:** 3

**Excitement:**

4: Strong: This paper deepens the understanding of some phenomenon or lowers the barriers to an existing research direction.

**Paper Topic And Main Contributions:**

This paper presents a novel task named Entity-Centric Question Generation, which aims to enhance topic-specific learning, assisted reading, and fact-checking processes. However, due to the absence of a task-specific dataset, the authors reconstruct the SQuAD 2.0 dataset with stricter constraints. In response to this challenge, they propose an innovative framework called GenCONE, comprising both upstream and downstream modules. The primary goal of GenCONE is to generate entity-centric questions with improved centricity, rationality, and answerability, thus addressing the limitations of existing question generation methods.

**Reasons To Accept:**

1. It proposes a novel task and reconstructed an existing dataset to task specific dataset. The constructed dataset may be useful.
2. It proposes a framework with both upstream and downstream module, while previous work only take one of this module into account.
3. Experiments show the good performance of the proposed framework.

**Reasons To Reject:**

This paper lacks of analysis of the constructed dataset. For example, it identifies central entity by rule according to section 4.1, some noise may introduce through this process but there is no sufficient discussion about this. Whats more, two constraints "corresponding questions contain and only contain the title entities" or "both context and question share and only share a common entity" is too strict since sentence with multiple entities is worthier for question in real application. But it's lack of statement of why must take these constraints.


**Reproducibility:**

4: Could mostly reproduce the results, but there may be some variation because of sample variance or minor variations in their interpretation of the protocol or method.

**Reviewer Confidence:**

4: Quite sure. I tried to check the important points carefully. It's unlikely, though conceivable, that I missed something that should affect my ratings.

---

> ### Author Rebuttal · Authors · 2023-08-29
>
> ## Response to Reviewer G3ne
>
> We would like to thank the reviewer for taking time to read our paper and thank them for their feedback.
>
> > This paper lacks of analysis of the constructed dataset. For example, it identifies central entity by rule according to section 4.1, some noise may introduce through this process but there is no sufficient discussion about this. Whats more, two constraints "corresponding questions contain and only contain the title entities" or "both context and question share and only share a common entity" is too strict since sentence with multiple entities is worthier for question in real application. But it's lack of statement of why must take these constraints.
>
> In dataset construction (Section 4.1), we applied these rules to reduce the noise introduced by entity extraction. These two constraints "corresponding questions contain and only contain the title entities" and "both context and question share and only share a common entity" are imposed to ensure that questions are tightly tied to a single key entity and more central to this entity. While multiple entities are more appealing in some scenarios, we start from a simple case where only one entity is specified. Relaxing the constraints could introduce ambiguity in evaluating the performance of our proposed GenCONE framework. We would also like to further investigate the multi-entity case in our future study.
>
> As per the reviewer’s suggestions, we manually analyzed the dataset we constructed. We consider a sample as good if the extracted entity is meaningful and the question is asking centered on the extracted entity, e.g., taking another entity, which is semantically related to the central entity, as the answer; and we consider a sample as bad if the question is asking something not exactly related to the extracted entity or the extracted entity is not a meaningful entity but rather a general word/phrase. We randomly selected 30 samples from the training set and 20 samples from the testing set, and show our results below.
>
> |                     | good        | bad       |
> | ------------------- | ----------- | --------- |
> | 30 training samples | 28 (93.33%) | 2 (6.67%) |
> | 20 testing samples  | 18 (90%)    | 2 (10%)   |
>
> Here are two examples of bad samples we found:
>
> ---
>
> - *Example 1*:
>
> Question: “What name was given to the plot to usurp power from the French House of Guise?”
>
> Entity: “Guise”
>
> ---
>
> - *Example 2*:
>
> Question: “Where are the Harvard medical, Dental and school of Public Health located?”
>
> Entity: “Public Health”
>
> ---
>
> There are commons in these bad samples: we should treat a longer phrase containing the extracted “entity” as a whole, such as “the French House of Guise” rather than “Guise” in Example 1, or “school of Public Health” rather than “Public Health” in Example 2, which is limited by the entity extraction tool we used. However, the percentage of good samples in the dataset is greater than 90%. We will also incorporate the above analysis in our revised version.

---

### Official Review · Reviewer_LJfA · 2023-08-05

**Soundness:** 4

**Excitement:**

3: Ambivalent: It has merits (e.g., it reports state-of-the-art results, the idea is nice), but there are key weaknesses (e.g., it describes incremental work), and it can significantly benefit from another round of revision. However, I won't object to accepting it if my co-reviewers champion it.

**Paper Topic And Main Contributions:**

This paper introduces a new task, Entity-Centric Question Generation (ECQG), which aims at generating questions asking an aspect of an entity. In the proposed GenCONE model, the Transformer decoder for QG is guided by focus locating and question verification modules, where the former learns what to ask, while the latter learns to ensure the answerability of question given the context.

Experiments on ECQG dataset derived from SQuAD illustrate that GenCONE notably outperforms both question-agnostic QG baselines and pretrained seq2seq models. The ablation study further demonstrates that the Transformer decoder benefits from both focus locating and question verification modules.

**Reasons To Accept:**

1. The paper is well-articulated, presenting a clear narrative from task definition to model architecture.

2. The proposed task of ECQG is beneficial for several applications, and a new  ECQG dataset is constructed.

3. The intuition behind the architecture is sound, where the answer-awareness of question generation is enhanced before and after the generation process.

4. Furthermore, the large performance gains of GenCONE over the baselines are impressive given the simplicity and applicability of its architecture.

**Reasons To Reject:**

1. The proposed methods are conventional and lack novelty.

2. While entity-centric question generation is of main interest in this paper, entity-centric properties of the model are less studied. For instance, the entities are simply prepended to the context to form the input, and it remains unclear whether this approach would allow question generation process to be centered on the entities.

3. In the model, focus locating and question verification modules both predict the same answer spans but with different types of input. Hence these two modules could be combined to reduce the computational cost.

4. In the ECQG dataset, the performance of large language models, such as GPT-3.5, should be provided as reference.

**Reproducibility:**

4: Could mostly reproduce the results, but there may be some variation because of sample variance or minor variations in their interpretation of the protocol or method.

**Reviewer Confidence:**

4: Quite sure. I tried to check the important points carefully. It's unlikely, though conceivable, that I missed something that should affect my ratings.

---

> ### Author Rebuttal · Authors · 2023-08-29
>
> ## Response to Reviewer LJfA
>
> We thank the reviewer for their valuable feedback and would like to address their concerns in the following:
>
> > 1. The proposed methods are conventional and lack novelty.
>
> While the individual components of our GenCONE model, such as content focusing and question verification, may draw from established techniques, it's important to emphasize that our contribution lies in their integration and adaptation to address ECQG. The novelty lies in the application of these components in a novel context rather than their isolated novelty, and each module is tailored to solve specific issues related to ECQG. We will emphasize this point more explicitly in our revised version.
>
> > 2. While entity-centric question generation is of main interest in this paper, entity-centric properties of the model are less studied. For instance, the entities are simply prepended to the context to form the input, and it remains unclear whether this approach would allow the question generation process to be centered on the entities.
>
> Entities play a crucial role in guiding the content focusing and question verification modules and all modules in our model work together to learn entity-context relation. We show with human evaluations (Section 5.4) that our proposed model outperforms baselines in centricity, which demonstrate the potential entity-centric property of our model. We will also delve deeper into this under-explored point in our future study, perhaps with entity embeddings or other mechanisms to better inform the model about the central entity.
>
> > 3. In the model, focus locating and question verification modules both predict the same answer spans but with different types of input. Hence these two modules could be combined to reduce the computational cost.
>
> The content focusing module captures the essential aspects of what to ask, while the question verification module ensures the answerability of the generated question. These modules serve different roles in enhancing the overall quality of question generation. While combining both might lead to some computational efficiency gains, the rationale for keeping them separate is to allow a degree of modularity and flexibility in refining each module. However, we will explore the option of their integration in future study to improve efficiency.
>
> > 4. In the ECQG dataset, the performance of large language models, such as GPT-3.5, should be provided as reference.
>
> We further evaluated the performance of GPT-3.5 on the ECQG dataset with zero-shot prompting. By providing prompt “Please ask an entity-centric question for entity <entity tokens> from the passage and give the corresponding answer: <passage tokens>”, we manually evaluated the outputs and compared them with ours / ground-truth. We show three examples below.
>
> ---
>
> - *Example 1*:
>
> [Entity] & Passage: [Lindzen's criticisms] & IPCC author Richard Lindzen has made a number of criticisms of the TAR. Among his criticisms, Lindzen has stated that the WGI Summary for Policymakers (SPM) does not faithfully summarize the full WGI report … John Houghton, who was a co-chair of TAR WGI, has responded to **Lindzen's criticisms** of the SPM …
>
> GPT-3.5: How did IPCC author Richard Lindzen's criticisms relate to the Third Assessment Report (TAR) and specifically to the WGI Summary for Policymakers (SPM)?
>
> GenCONE: Who has responded to Lindzen's criticisms of the WGI?
>
> Ground-truth: Who responded to Lindzen's criticisms?
>
> ---
>
> - *Example 2*:
>
> [Entity] & Passage: [cervical cancer] & Another important role of the immune system is to identify and eliminate tumors. This is called immune surveillance. The transformed cells of tumors express antigens that are not found on normal cells. To the immune system, these antigens appear foreign, and their presence causes immune cells to attack the transformed tumor cells. The antigens expressed by tumors have several sources; some are derived from oncogenic viruses like human papillomavirus, which causes **cervical cancer** …
>
> GPT-3.5: How does the immune system recognize and respond to cervical cancer, and what is one of the sources of antigens associated with this type of cancer?
>
> GenCONE: What virus causes cervical cancer?
>
> Ground-truth: What is the virus in humans that causes cervical cancer?
>
> ---
>
> - *Example 3*:
>
> [Entity] & Passage: [X.25 networks] & There were two kinds of **X.25 networks**. Some such as DATAPAC and TRANSPAC were initially implemented with an X.25 external interface. Some older networks such as TELENET and TYMNET were modified to provide a X.25 host interface in addition to older host connection schemes … X.75 and X.121 allowed the interconnection of national **X.25 networks** …
>
> GPT-3.5: How did the development of X.25 networks differ between various implementations, and how did X.75 and X.121 contribute to the interconnection of these networks?
>
> GenCONE: What were the two types of X.25 networks?
>
> Ground-truth: How many types of X.25 networks were there originally?
>
> ---
>
> In our analysis, we observed that GPT-3.5 tends to generate more intricate and open-ended questions compared with those generated by GenCONE or found in ground truth. Unlike the factoid questions predominant in our dataset, the questions generated by GPT-3.5 often require a more nuanced understanding of the context and typically cannot be answered by a text span of the input passage. Moreover, while some questions generated by GPT-3.5 can be answered within the given context, others extend beyond it, requiring additional information. Therefore, although GPT-3.5 can still generate questions that focus on specific entities and appear rational, GenCONE demonstrates superior performance in terms of answerability. We will also involve the above analysis in our revised version.

---

### Meta-Review · Area_Chair_QZRy · 2023-09-19

**Recommendation:** 3

**Metareview:**

**Summary:** The paper proposes a new task: Entity-Centric Question Generation (ECQG) which aims at generating questions pertaining to specifically to an entity of interest (EOI). This is done using two modules: focus locating (to identify EOI), and a question verification module (to evaluate answerability of generated questions). Due to the lack of a task-specific dataset, the paper creates a dataset starting from SQuAD 2.0. Empirical and human evaluation shows that the proposed ECQG approach: GeneCONE outperforms both question-agnostic QG  and pretrained seq2seq baselines.

**Approach and Methodology:** In the initial reviews some reviewers expressed concern on the limited novelty of the proposed solution, and the task being inspired from a previous work . In my opinion, the task setup is sufficiently distinct from previous works (with some possible real world applications) with the key novelty of the approach being restricted to plugging together existing solutions from previous ideas (focus locating and question verification) to get an end-to-end pipeline for entity-centric QG.

**Empirical Evaluation:** The quantitative empirical evaluation in the paper is sound, with appropriate comparisons with baselines when needed (and shows that the approach beats the baselines). The dataset constructed in this paper may be an useful to the community. The main concerns regarding experiments is the limited human evaluation and discussions/qualitative studies performed to highlight some of the core motivations for the approach: centricity of generated questions on the entities, coherence evaluation of QA pairs, etc. The paper contains a small human evaluation study (over 100 samples) which needs to be made more extensive to strengthen the paper.

**Recommendations for Improvement:** (i) Expand human evaluation study to make it more extensive, and clearly highlight analysis specific to entity-centric nature of GeneCone and improvements over baselines.

(i) Add quantitative results (in addition to the qualitative results provided in author response) for benchmarking ECQG dataset with LLMs like ChatGPT, GPT-4, etc.

---

### Decision · Program_Chairs · 2023-10-07

**Decision:**

Accept-Findings

**Comment:**

**Summary:** The paper proposes a new task: Entity-Centric Question Generation (ECQG) which aims at generating questions pertaining to specifically to an entity of interest (EOI). This is done using two modules: focus locating (to identify EOI), and a question verification module (to evaluate answerability of generated questions). Due to the lack of a task-specific dataset, the paper creates a dataset starting from SQuAD 2.0. Empirical and human evaluation shows that the proposed ECQG approach: GeneCONE outperforms both question-agnostic QG  and pretrained seq2seq baselines.

**Approach and Methodology:** In the initial reviews some reviewers expressed concern on the limited novelty of the proposed solution, and the task being inspired from a previous work . In my opinion, the task setup is sufficiently distinct from previous works (with some possible real world applications) with the key novelty of the approach being restricted to plugging together existing solutions from previous ideas (focus locating and question verification) to get an end-to-end pipeline for entity-centric QG.

**Empirical Evaluation:** The quantitative empirical evaluation in the paper is sound, with appropriate comparisons with baselines when needed (and shows that the approach beats the baselines). The dataset constructed in this paper may be an useful to the community. The main concerns regarding experiments is the limited human evaluation and discussions/qualitative studies performed to highlight some of the core motivations for the approach: centricity of generated questions on the entities, coherence evaluation of QA pairs, etc. The paper contains a small human evaluation study (over 100 samples) which needs to be made more extensive to strengthen the paper.

**Recommendations for Improvement:** (i) Expand human evaluation study to make it more extensive, and clearly highlight analysis specific to entity-centric nature of GeneCone and improvements over baselines.

(i) Add quantitative results (in addition to the qualitative results provided in author response) for benchmarking ECQG dataset with LLMs like ChatGPT, GPT-4, etc.